# The Feasibility of Implementing the Flexible Surge Capacity Concept in Bangkok: Willing Participants and Educational Gaps

**DOI:** 10.3390/ijerph18157793

**Published:** 2021-07-22

**Authors:** Phatthranit Phattharapornjaroen, Viktor Glantz, Eric Carlström, Lina Dahlén Holmqvist, Yuwares Sittichanbuncha, Amir Khorram-Manesh

**Affiliations:** 1Institute of Clinical Sciences, Department of Surgery, Sahlgrenska Academy, Gothenburg University, 40530 Gothenburg, Sweden; amir.khorram-manesh@surgery.gu.se; 2Department of Emergency Medicine, Faculty of Medicine, Ramathibodi Hospital, Mahidol University, Bangkok 10400, Thailand; raysc.yuwares@gmail.com; 3Trauma Unit, Department of Surgery, Sahlgrenska University Hospital, 40530 Gothenburg, Sweden; viktor.glantz@vgregion.se; 4Institute of Healthcare Sciences, Sahlgrenska Academy, Gothenburg University, 40100 Gothenburg, Sweden; eric.carlstrom@gu.se; 5USN School of Business, University of South-Eastern Norway, P.O. Box 235, 3603 Kongsberg, Norway; 6Department of Internal Medicine and Clinical Nutrition, Institute of Medicine, Sahlgrenska University Hospital, 40530 Gothenburg, Sweden; lina.holmqvist@vgregion.se; 7Department of Research and Development, The Swedish Armed Forces Center for Defense Medicine, Västra Frölunda, 42676 Gothenburg, Sweden

**Keywords:** alternative care facilities, disasters, flexible surge capacity, major incidents and disasters, surge capacity

## Abstract

The management of emergencies consists of a chain of actions with the support of staff, stuff, structure, and system, i.e., surge capacity. However, whenever the needs exceed the present resources, there should be flexibility in the system to employ other resources within communities, i.e., flexible surge capacity (FSC). This study aimed to investigate the possibility of creating alternative care facilities (ACFs) to relieve hospitals in Bangkok, Thailand. Using a Swedish questionnaire, quantitative data were compiled from facilities of interest and were completed with qualitative data obtained from interviews with key informants. Increasing interest to take part in a FSC system was identified among those interviewed. All medical facilities indicated an interest in offering minor treatments, while a select few expressed interest in offering psychosocial support or patient stabilization before transport to major hospitals and minor operations. The non-medical facilities interviewed proposed to serve food and provide spaces for the housing of victims. The lack of knowledge and scarcity of medical instruments and materials were some of the barriers to implementing the FSC response system. Despite some shortcomings, FSC seems to be applicable in Thailand. There is a need for educational initiatives, as well as a financial contingency to grant the sustainability of FSC.

## 1. Introduction

The rate of major incidents and disasters (MIDs), irrespective of the causes, has gradually risen over the past two decades. A major proportion of these incidents are triggered by natural hazards as a result of climate changes and can result in potentially deadly consequences [1]. MIDs can result in overwhelming numbers of physical and mental injuries, and lead to socioeconomic challenges, which can surpass healthcare response capability and capacity [2,3,4,5]. The most significant goal of the healthcare system during a MID is to provide care to victims and minimize their suffering by using available resources. The emergency management organization has to facilitate preparedness and relief measures to create a well-organized contingency plan. However, the ability to manage surge capacity is central to consolidate and optimize the system.

Surge capacity consists of four essential elements: staff, stuff (devices), structure (spaces), and system. All elements focus on three levels of healthcare operation; public-based, hospital-based, and community-based [6,7,8]. Each level has its capabilities and limitations regarding surge capacity in all phases of an emergency. In a critical situation, hospitals will try reallocating patients or medical equipment (like ventilators), conducting primary and secondary surge capacity, but the overflow of patients might still outstrip their ability and result in unpredictable consequences, as illustrated in northern Italy during the COVID-19 pandemic in 2020 [9]. Therefore, the amplification of MID consequences requires another effort of surge capacity, i.e., “flexible surge capacity” (FSC). FSC aims at scaling up and down all viable resources in the community in terms of all four elements of surge capacity mentioned above [10,11,12]. As an example, in a previous study, using emergency physicians as alternative leadership in the management of MIDs was discussed within the concept of FSC [10,13]. Other reports have also pointed out the importance of systems and rules in MID management [14]. However, there are limited studies investigating the use of Alternative Care Facilities (ACFs), which are places that can potentially be modified into treatment stations for disaster-related patients or non-disaster patients, to incorporate the structural needs during MIDs as the FSC [15,16]. The concept of FSC and the use of ACFs were found to be feasible in Sweden, but it may be unfeasible in other countries with varied structures and cultures [10,11]. Nevertheless, the current COVID-19 pandemic demonstrates a global concern about ACFs and forced many healthcare organizations to adapt schools and private clinics into isolation or vaccination areas, while others either rapidly modified some parts of the hospitals or one small hospital for critical and pandemic care [15,16,17].

Thailand has experienced several MIDs (e.g., tsunamis and terrorism) [2], and difficulties regarding structure and system elements of surge capacity during crises. There have been efforts to enhance multi-agency partnerships like civil and military collaborations to strengthen the MID management system. However, there is no consensus regarding organized collaboration routines and procedures. The accomplishment of FSC may provide an opportunity to utilize institutional, governmental, and private actors’ resources to achieve routines and consensus about the multi-agency management of emergencies. Such an achievement may provide a model for low and middle-income countries, especially in Asia. This study aimed to investigate the feasibility of implementing FSC response systems by examining the needs for, and the possibility of using ACFs, as well as examining any potential barriers to them.

## 2. Materials and Methods

Venue: For this study, ACFs in Bangkok, Thailand, including public primary healthcare, private, dental, and veterinary clinics, schools, sports arenas, and hotels, were investigated. The reason for choosing Bangkok was the existing variability and the substantial number of potential facilities in the city.

Sample: The names and locations of public primary healthcare centers, private clinics, and dental clinics were obtained from the Ministry of Health in Bangkok. The names of schools were obtained from the Department of Education. Veterinary clinics, hotels, and sports arenas were searched for online through available websites. The sampling for government-related institutions was conducted by the Ministry of Health and Department of Education officials, and the authors had no influence on the selection. For other facilities, all available facilities were contacted and received the questionnaire. However, the authors had no influence on the response from these facilities, and despite multiple contacts, only those with complete responses were included.

All aforementioned facilities received the questionnaire. The ministries, department, and responsible persons in each facility were contacted by the main author to decide whether the questionnaire should be distributed centrally or by sending to each individual entity.

The Questionnaire (Appendix A): An already published and validated questionnaire (Cronbach’s alpha with an internal consistency of 0.739 [18]) was utilized [11]. It was translated into Thai and then back into English to assure the accuracy of the questions. The face validation of the original questionnaire was based on logic, relevance, comprehension, legibility, clarity, usability, and consensus. The questionnaires referred to a situation that the facilities of interest faced, a fictitious scenario of a mass casualty incident. ACFs were asked about the care they could provide to the healthcare actors in the area.

There were both open-ended and close-ended questions to capture and generate relevant data, and the answers were quantitatively collected [19,20]. The quantitative data dealt with the number of participants, who received the questionnaire and the number of those responding, and, thus, the rate of participation. No other quantitative data were presented.

The qualitative data were collected by semi-constructive interviews, which in contrast to a structured interview technique, allows the participant to divert and suggest new ideas during the interview based on responses. The interviewer in a semi-structured interview generally has a framework of themes to be explored [21]. All data were recorded and analyzed by the first investigator.

The theme applied to the interview was the one used in Glantz’s study (2020) [11], which used the same questionnaire and interview questions. For analyzing, we deductively used the distribution of the concepts of surge capacity, which allows for examining communication by using text directly. This method allows for both qualitative and quantitative analysis if needed. It is also considered a relatively exact research method if it is done correctly (limitation). It is an inexpensive research method and considered a more powerful tool when combined with other research methods such as interviews. However, it also has its limitations (see limitation). Thematic content-coding was performed to identify competencies, challenges, and interest to take part in the FSC response system. No more interviews were held after reaching the point of data saturation [20].

## 3. Results

From the sample of 967 names and addresses of the facilities, 228 responses (23.6%), were collected through Google forms. Participants who replied to the questionnaire answered all of the questions. However, the response rate varied in the different groups (Appendix A).

The highest response rate was the public primary healthcare centers (PHCC) at 50.7%. Only 13 out of 185 private clinics responded (7%) (Table 1). Additionally, municipal schools with less than 500 students were excluded since they were considered to be too small to provide help in the FSC response system. The absolute number of facilities and respondents after exclusion was 739 and 162 (21.9%), respectively.

After responses were returned from potential ACFs, the phone interviews were conducted with directors, owners, and administrators of the facilities. Face-to-face interviews were not attainable due to the social distancing policies and the shutdown of public services to mitigate the COVID-19 pandemic. The questions were discussed in-depth during 15–30 min interviews with directors from ten primary health care centers, six private clinics, two dental clinics, two veterinary clinics, one school director, one sports facility administrator, and one hotel owner.

### 3.1. General Results

The majority of alternative care services voluntarily offered their facilities in the event of a MID. A considerable number of both public primary healthcare and dental clinics proposed to manage patients with minor injuries, manage mild medical conditions, provide psychosocial support, communicate to emergency medical services, and transport severely injured patients to hospitals. Interviewees from these facilities raised concerns over unproportioned staff supply, physical space, medical equipment, and material resources. Veterinary clinics, schools, and sports halls were interested in providing their particular facility to house an ACF. All facilities showed an interest in taking part in educational initiatives such as training in first aid, cardiopulmonary resuscitation (CPR), major trauma care, and the transportation of victims. Some facilities reported the benefits to specifically enhance their performance in MID management. Nevertheless, the shortage of medical supplies was one of the apprehensive barriers of non-healthcare facilities’ administrators.

### 3.2. ACF Specific Results

#### 3.2.1. Health Care Clinic

Health care clinics in Thailand exist within two particular systems. Firstly, the public primary healthcare centers (PHCC), governed by the Ministry of Health, have doctors, nurses, and pharmacists. The PHCCs provide routine treatments for patients with chronic diseases and minorly acute illnesses. Private clinics, which operate independently, yet require a license from the Ministry of Health, have a doctor, and either a nurse or assistant nurse. Most private clinics are specialty clinics providing care for specific niches like pediatrics allergies or dermatologic issues. Thirty-five out of sixty-nine public primary healthcare centers and thirteen out of one-hundred-and-eighty-five private clinics responded to the questionnaire. A substantial number of respondents possessed the capability to serve in the FSC response system by treating fewer injured people and providing psychosocial support to patients and staff. The public centers would prefer to stabilize patients physiologically before transport to a major hospital. Furthermore, a limited number of private clinics offered their staff and stuff to either treat at their clinics or move their personnel to affected facilities. Two of them offered no potential to be included in an FSC response system (Table 2).

In addition, more than half of the respondents (twenty-one public centers and ten private clinics) commented that they lacked essential medical equipment including advanced defibrillators, intubation equipment, and resuscitation medications. Moreover, increased space, extra ambulance vehicles, and more emergency kits would ease and encourage their contribution. The in-depth interviews highlighted this insufficiency in both public and private healthcare facilities. Although all public facilities had doctors and nurses, and some centers had pharmacists to include in a potential FSC, the qualitative data collection revealed barriers in private clinics’ willingness to contribute doctors or nursing assistants. Furthermore, many respondents requested training in advanced life support, emergency and trauma case management, disaster management, and prehospital transportation in their interviews. Though most respondents offered constructive criticism, a limited number of answers (three public clinics and five private clinics) responded negatively toward any kind of involvement.

#### 3.2.2. Dental Clinics

All dental clinics in Bangkok represented private practices and provided a wide range of dental procedures. From the one-hundred-and-sixteen clinics contacted, seventeen responses were received. Sixteen out of seventeen clinics responded they could provide care to patients with minor injuries in the event of a MID. Approximately one-third of clinics were willing to offer psychosocial support to patients and staff. Several expressed a willingness to perform minor surgical procedures and treat other acute cases to relieve the hospital emergency department. Additionally, one dental clinic offered space, instruments, and materials, and another expressed the potential to stabilize a patient before transfer to a major hospital. Ten clinics lacked equipment and devices, such as automated external defibrillators (AEDs), monitor sets, medical oxygen tanks, and splinting material. Additionally, all clinics commented they would voluntarily support FSCs if they received more education in emergency care. Two clinic owners interviewed expressed confidence in managing all dental injuries and minor wound care, but not overall MID management (Table 2).

#### 3.2.3. Veterinary Clinics

All veterinary clinics were private actors. We received fourteen responses from the ninety contacted facilities, and all respondents reported an ability to participate in FSC. All of them indicated that they would be able to offer treatment for minor injuries from the incident. One clinic expressed the ability to perform minor surgery and provide wound treatment. Another clinic indicated that it could offer support for psychosocial issues as part of an FSC system. Two of them had physical space which could be utilized to provide any kind of care in MIDs. Most of the clinics were concerned about their jurisdiction in human injury management (Table 2). It was perceived from the forms and interviews that veterinarians thought they were obligated and permitted to handle only animals. Nonetheless, three out of fourteen clinics were interested in improving their knowledge of human life-saving procedures.

#### 3.2.4. Schools

Only municipal schools from the Ministry of Education were contacted in this study. A sum of 136 out of 437 replied from schools with a student population ranging from 100 to more than 2000. Because of the requirements for FSC, only institutions with significant staff, space larger than 3000 square meters, and with more than 60 employees were included from the entire set of respondents (*n* = 70). Absolute numbers of respondents comprised 70 schools, of which 23 reported their abilities to stop bleeding, repair wounds, administer CPR, and perform emergency procedures. Half of the participants expressed a willingness to manage minor injuries and offered care for children. All institutions had a small treatment room to look after sick or injured staff as well as students. The room displayed medical equipment such as first aid kits, blood pressure cuffs, and thermometers. A few schools identified it as a relevant resource to the collaboration with nearby primary care centers and hospitals in case of emergencies. Almost all respondents voiced that they demonstrated no readiness to manage the situation because of their shortage in manpower and skills. However, 23 of the respondents communicated their enthusiasm for participating in training drills in resuscitation and emergency care to be prepared for daily student injuries and incidents. Additionally, one of the institutions proposed a segment for medical treatment to be included in their academic curriculum, while another preferred to generate their own MID management plan (Table 3).

#### 3.2.5. Sports Facilities

All sports centers were municipally operated and had a small medical clinic with either nurses or assistant nurses who were trained in first aid and wound care. Five out of twelve sports centers returned questionnaires, and the forms were sufficiently answered by the directors of the facilities. All respondents were interested in the FSC response system and reported to be capable of managing minor injuries. Two of them could stop major hemorrhages and repair wounds, and one of them could offer homeless or affected people accommodation as well as psychosocial support. Three out of five sports centers reported being uncomfortable when confronted by MIDs due to a lack of manpower and material (Table 3).

#### 3.2.6. Hotels

Eight out of fifty-eight questionnaires were returned. All of them showed an interest in participating in the FSC response system as temporary housing facilities and sources of water and food for people concerned. Three hotels reported their potential to manage minor injuries. Four hotels offered childcare assistance to parents that might be needed during the activation of an FSC system. One of the hotels reported that staff were educated with annual first aid courses. Nevertheless, all hotels expressed shortcomings in educational initiatives and medical supplies (Table 3).

## 4. Discussion

The most interesting outcome of this study was the increased awareness of participants and authorities involved in the need for a FSC system. The involvement of authorities allowed for the conduction of the study in both disseminating the questionnaire and returning respondents’ comments. Results also demonstrate the willingness of carrying out FSC in Bangkok among participants. MID management demands a multi-agency approach. From this perspective, this study also indicates different organizations may be able to collaborate to achieve the concept of FSC. However, there are some requirements that should be fulfilled to achieve a successful FSC [10,11,22,23,24]. Staff, stuff, structure, and system (4S) remain the significant elements of the FSC response system. Previous researchers have identified the necessity of surging capacity during MIDs [6,25,26]. FSC aims to activate other resources that are not usually considered in contingency plans. These resources can be generated, as shown in this study, within a community by using its own pool of resources. As noted by the World Health Organization (WHO), this process is part of striving to reach the Sustainable Development Goal 11 of developing community resilience [27]. The provision of alternative central or local leadership by offering command and control to the public health agency and emergency physicians, respectively, has already been introduced [10,13]. There has been a discussion in the literature regarding ACFs and their importance [15,16]. This paper aimed to highlight the possibilities for ACFs in the metropolitan area in an emergency-prone country [28]. The knowledge of staff, stuff, and structure enables the development of necessary rules, regulations, and systems.

The maximization of the potential ACFs, together with a well-organized system, can significantly improve the survival of victims. In addition, the key to initiate change and advance the system forward is not only the availability of resources but also the willingness of organizations to collaborate. The results of this study positively display the willingness of facilities to alleviate hospitals and partake in a response to a MID. Although the number of responses was limited, the study showed that the majority of participants were willing to alleviate the burdens of MIDs. These participants represent the facilities’ leadership. The choice of respondents is based on the assumption that the leadership has an overview of the organizational capacities and the ability to act as the voice of the staff [29]. The four components of surge capacity will be discussed below.

### 4.1. Staff

Not only the quantity of personnel but also the quality of manpower is one of the essential elements in the FSC response system. Competent workforces influence disaster management and reflect a well-structured incident command system. Staff should possess appropriate credentials and recognize their roles and their facility’s disaster response plan. Furthermore, proficient leadership is critical to promote a more superior command and control of the staff and result in efficient MID response [13]. This study showed a significant number of staff in the investigated facilities of interest were willing to partake in the response system. Nevertheless, they raised concerns regarding knowledge, confidence, and discomfort in procedural techniques. The results of several studies are congruent with the one in this study. Both healthcare and non-healthcare personnel volunteered as emergency responders in case they received proper training [10,24,30,31]. This result differs from a Swedish study in which non-healthcare staff were unwilling to partake in MID management due to a lack of competency [11]. To resolve the problem with the lack of competency among staff, short-term recommendations are to establish a dynamic personnel base and staff pooling across facilities, or recruit solicited employees like retirees, as well as unsolicited employees [6,7]. In a long-term development, the establishment of first aid, emergency procedures, patient transfer, and psychosocial support courses with annual re-certification would represent a suitable strategy to empower staff to participate in the FSC response system [11,30,32].

### 4.2. Stuff (Health Care Equipment and Materials)

The scarcity of medical equipment and materials during MIDs in the past had a global impact [8,11,24,26,33]. The facilities of interest investigated in this study are proportionally unequipped to serve victims with severe injuries, which in turn means that their stockpiles and their capacities to replenish them are limited. The investigated facilities were unprepared for mass casualty events, although they showed a willingness, capacity, and capability of doing so [11]. Many also expressed a need for qualified resources in order to be able to partake in a FSC response. Moreover, financial stability and contingency funding, along with a rapid acquisition and distribution system of medical instruments and materials, should be discussed beforehand to prevent any depletion of resources [8,25,26]. The reallocation of medical instruments and devices to both public and private actors is one of the crucial steps in MID management that also need collaboration and coordination among facilities [25].

### 4.3. Structure

Most of the investigated facilities had their own treatment areas, including sports centers and schools, however, the sectors were initially designed to fit a compact group of people at any one time. A rearrangement of instruments, devices, and materials is needed to increase utilizable spaces like patient bed areas, treatment areas, counselling areas, and operation areas. Primary healthcare, dental, and veterinary clinics occupied more suitable spaces, as some of them already utilized a sterile area for minor operations, and their whole floor was often structured for the provision of medical care. Therefore, they demanded only minimal retrofitting and they would be more equipped to handle the situation than non-healthcare facilities such as sports centers, schools, and hotels [11]. Despite the limited treatment area in sports facilities and schools, they possess enormous vacant fields that can act as an area to construct novel nursing zones for victims. In addition, a civil and military collaboration could organize and manage a field hospital and military transportation which could be another one of the options to utilize schools and sports arenas [24]. Moreover, one example of a recent MID was the first wave of COVID-19 pandemic in 2019 that triggered the surge capacity of the structures needed in many countries to alleviate the burden of the major hospitals [24,33]. The event justifies the concept of a FSC response system, however, the efforts need to be drilled, trained, and practiced [10,11,34].

### 4.4. System

In Thailand, there was an outstanding difference between public and private organizations. While more than 30% of governmental institutions declared their intention to assist, only 11% of the private actors answered the forms, which indicated a lack of interest in the issues, and they have an unclear line of command because they are independently governed, and the Ministry of Health has no power over them. One possible reason for the lack of interest from private actors can be the differences in financing. While public organizations are financed by taxes, private actors have no government funding for disaster preparedness and humanitarian work. Consequently, regulations are needed to secure participation from all types of actors. Nevertheless, when a MID occurs, the need for involvement by all relevant actors is unavoidable. Therefore, the private sector should establish a command system that corresponds with the national system. The collaboration between the private and public sectors would advance with the urge of the Ministry of Health to exhibit the corresponding guidelines and protocols [35,36,37]. Although the Swedish results for differences between public and private actors were inconclusive, the governing system in Sweden offers municipal and community independence, which means that government can only recommend necessary measures [11].

To create a robust MID management system, multi-agency collaboration regarding surge capacity and the systematic organization of resources is recommended. Inter-agency partnership cannot be a spontaneous action or initiative, as it is a challenge to perform smooth communication and co-operation during MIDs. Therefore, exercises, training, and practice guidelines may empower both public and private organizations to learn how to work together in emergencies like hospital evacuations [2,13,14,34,38,39].

## 5. Limitation

The results of this study are from one urban Asian society. A similar study in a non-urban society could lead to a different outcome. Despite several efforts, the response rate in this study was low. One reason for the low participation rate might be the perception of disasters as being rare events, as the awareness of MIDs is low among the Thai population, government officials, and business stakeholders. Another reason for low participation might be the fact that disaster management is based on both medical and non-medical measures. There might be a limitation in understanding between organizations involved, i.e., non-medical institutions having limited knowledge about the limitations and possibilities of medical institutions and vice versa [24]. In this study, top authorities were contacted to improve the response rate with low success. However, the response rate is concordant with previous non-pandemic studies. In future studies, electronic repeated reminders might improve response rates [40,41,42].

Finally, the content analysis is subject to error, particularly when a relational analysis is used to attain a higher level of interpretation. Nevertheless, the main investigator was part of a similar study published in 2020 and therefore had prior experience with the analytical method.

## 6. Conclusions

The concept of a FSC response system is applicable to the metropolitan area of this study where several clinics, schools, and hotels exist. New studies will reveal if the apprehensive points could be generalized to other countries with the same context. However, the development and implementation of educational initiatives including exercises, drills, maintenance courses, and financial contingency were recognized to be barriers necessary to overcome in the implementation of the FSC concept. The success of overcoming these barriers would enable sustainable FSC systems to be developed within the MID response system.

## Figures and Tables

**Table 1 ijerph-18-07793-t001:** The response rate of all ACFs in Bangkok.

Alternative Care Facilities	Number of Facilities	Number of Response (%)
Total	967	228 (23.6)
Public primary health care centers	69	35 (50.7)
Private clinics	185	13 (7)
Dental clinics	116	17 (14.6)
Veterinary clinics	90	14 (15.6)
Schools	437	136 (31.1)
Sport facilities	12	5 (41.7)
Hotels	58	8 (13.8)

**Table 2 ijerph-18-07793-t002:** Questionnaire data of potential support that clinics could provide.

Choices of Provided Facilities	PHCC (35)	Private Clinic (13)	Dental Clinics (17)	Veterinary Clinics (14)
Fewer injured patients from the incident	30	10	16	14
Stabilization of seriously injured patient before transfer to major hospital	12	1	1	1
Other hospital emergencies	6	1	2	0
Assisting the hospital	6	1	3	0
Resources; space, instruments, materials	6	1	1	2
Minor procedures	5	0	2	1
Medical patients	10	0	1	0
Psychosocial support to patients and staff	21	3	5	1
Coordination of home transportation	10	3	3	0
Cannot help	1	2	0	0

**Table 3 ijerph-18-07793-t003:** Questionnaire data of potential support those facilities could provide.

Choices of Provided Facilities	School (70)	Sport Facilities (5)	Hotel (8)
Stop bleeding, wound management, or emergency procedures	23	2	1
Minor injury	35	5	3
Psychosocial support to patients and staff	25	1	2
Shelter for homeless or injured people	8	1	8
Food and water	30	0	8
Childcare for health care staff	35	1	4
Share own staff with other organization	10	0	0
Cannot help	0	0	0

## Data Availability

Datasets used and analyzed during the current study are available from the corresponding author upon reasonable request.

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
