# Peer review of "The Feasibility of Implementing the Flexible Surge Capacity Concept in Bangkok: Willing Participants and Educational Gaps"

_ijerph, 2021, doi:10.3390/ijerph18157793_

Round 1
Reviewer 1 Report
Good attempt in writing about surge capacities and needed coordination with multiple elements involved in it. Very appropriate for current situation.
Author Response
Thank you very much for your time dedicated to review the manuscript and your comment.

Reviewer 2 Report
I find the manuscript fine in its present form.
Author Response
We also seized the opportunity to go over the manuscript for brevity, clarity, and consistency, allowing a native English speaker academic perform the necessary English revision.
Thank you very much for your time dedicated to review the manuscript and your comment.

Reviewer 3 Report
The article deals with an important topic that is gaining importance in times of globalization and crises, such as the current pandemic.
However, in my opinion, the research carried out here is of little importance (both for science and for practical applications), and the final conclusions were drawn incorrectly.
The article requires the following corrections:
1. The title should be changed to describe the actual content of the article - a) the research does not concern Thailand, but Bangkok b) in my opinion, the collected research material is too modest to draw conclusions about the feasibility of FSC implementation because it depends on many factors other than just the answers respondents. So, the title of the article should contain two words: Bangkok and survey.
2. In Table 2, the numbers do not sum up to the Total value, which raises doubts as to the reliability and whether the authors have lost control of the experiment. Taking this into account, the raw research data should be published. As the polls were collected using a google form, they can be easily exported to csv and uploaded to any public repository - e.g. git or zenodo. Also, structured notes during semi-structured interviews should be published. I assume that access to this data could be useful for scientists dealing with similar topics.
3. The discussion chapter should focus more on the discussion of the results obtained than on speculation and review of the literature.
4. The chapter on the limitation of the research is inconsistent with the chapter on conclusions. The results obtained with such limited studies cannot be generalized to the entire country, and certainly not to other countries. So the Conclusions chapter should be rewritten anew.
Author Response
Response to Reviewer 3 Comments
Reviewer: The article deals with an important topic that is gaining importance in times of globalization and crises, such as the current pandemic. However, in my opinion, the research carried out here is of little importance (both for science and for practical applications), and the final conclusions were drawn incorrectly.
Response: Thank you for your comment. As you mention the topic is important and, as far as we know, no one has before done this kind of research. So, even if the results might not be conclusive but hopefully this paper will initiate a discussion and lead to complementary studies.
We also seized the opportunity to go over the manuscript for brevity, clarity, and consistency, allowing a native English speaker academic perform the necessary English revision.
The article requires the following corrections:
Point 1 The title should be changed to describe the actual content of the article - a) the research does not concern Thailand, but Bangkok b) in my opinion, the collected research material is too modest to draw conclusions about the feasibility of FSC implementation because it depends on many factors other than just the answers respondents. So, the title of the article should contain two words: Bangkok and survey.
Response 1: We appreciate your comments on the title and change it to a more suitable one. We also agree that implementation of any measure depends on many factors but would like to emphasize that FSC has been defined as extra resources within the community that can be activated and as you mention it also depends on the good will of the community members but of course needs educational initiative, as stated in the new title.
Point 2 In Table 2, the numbers do not sum up to the Total value, which raises doubts as to the reliability and whether the authors have lost control of the experiment. Taking this into account, the raw research data should be published. As the polls were collected using a google form, they can be easily exported to csv and uploaded to any public repository - e.g. git or zenodo. Also, structured notes during semi-structured interviews should be published. I assume that access to this data could be useful for scientists dealing with similar topics.
Response 2: Thank you for your suggestions. The reason why the numbers in table 2 did not match with the total value is that the total values are the numbers of respondents and not the quantities of the answers. This is clarified in the new version.
The facilities of interest had the opportunities to describe their capabilities and many of them were willing to participate, for example, one clinic could provide care for less injured patients, stabilizing serious injured patients before transfer to major hospital, while another clinic would only assist the hospitals, performing minor procedures, and treating medical patients at the same time.
All material is in Thai but are attached as appendix B.
Point 3 The discussion chapter should focus more on the discussion of the results obtained than on speculation and review of the literature.
Response 3: Thank you for your comments. In the new version we have clarified that the results are discussed in the light of the literature.
Point 4 The chapter on the limitation of the research is inconsistent with the chapter on conclusions. The results obtained with such limited studies cannot be generalized to the entire country, and certainly not to other countries. So the Conclusions chapter should be rewritten anew.
Response 4: We see the reviewers’ points about inconsistency, and we rewrote the conclusion as followed. (Changed in the manuscript line 407-415). However, we would also emphasize that a pilot study has already been done in Sweden with similar results and thus may indicate that the concept can be used in other countries.
The concept of an FSC response system is applicable to the metropolitan area of this study where several clinics, schools and hotels exist. New studies will reveal if the apprehensive points could be generalized to other countries with the same context. However, the development and implementation of educational initiatives including, exercises, drills, and maintenance courses and financial contingency were recognized to be barrier necessary to overcome in implementation of the FSC concept. The success of overcoming these barriers would enable sustainable FSC systems to be developed within the MID response system.

Reviewer 4 Report
The paper details the feasibility of implementing flexible surge capacity by creating 'Alternative Care Facilities' in Bangkok, Thailand. It is written by a somewhat curious mix of researchers from Norway, Sweden and Thailand. The subject is very relevant in 2021.
Generally, the paper is well-written but could do with a final spell check, I noticed some minor style errors.
I have some comments which I will outline below, noting the line number in my review document:
37 even though you note 'irrespective of the causes', the majority of incidents reported in the Red Cross report seem to stem from climate change. You could address this, since it's good explanation for the gradual rise in disasters.
53 and onward - the description of FSC could be structured more, it is not entirely clear now and some sentences seem to be placed here haphazardly, for example ' In previous studies, alternative leadership within the concept of FSC has been discussed(10,13).' does not logically flow in the narrative.
57 when introducing ACF please describe in more detail what this is. Are they physical sites? They are described like they are but it's not entirely clear to the reader.
84 It is not clear to me what you did with the 'Other facilities of interest; veterinary clinics, hotels, and sport facilities'. Did you find these on Google and send them a questionnaire? In line 112 you mention excluding schools with less than 500 students, did you have a cutoff for these other categories?
87 to who at the ACF was this questionnaire sent? The answers could be very different based on the function of whoever fills in this questionnaire.
115 the description of completion is unclear - do you mean that of those that filled in (e.g. 13.8 of hotels) they only filled in half the questionnaires?
**Starting at 3.2 ACF specific results: please start the subheading with a short description of how these facilities work in Thailand/BKK, I am a bit confused as to why some of the amounts are so low (please see below).
155 I am not sure if I entirely understand the scores: does this mean that only 12 out of 35 'public primary health care center' felt they could 'stabilize serious injured patients' ? How are these centers organized? Why can only a few 'primary health care center' actually take care of patients? Considering the name I would expect more capabilities. The same goes for private clinics.
**Discussion: the discussion is good and clearly written.
331 (5. Limitations): obviously there is a huge difference between a city the size of my country and 'non-urban settings'. Could you think of more limitations? Also, do you have any ideas as to why response rate was low? How could you improve this?
Author Response
Reviewer: The paper details the feasibility of implementing flexible surge capacity by creating 'Alternative Care Facilities' in Bangkok, Thailand. It is written by a somewhat curious mix of researchers from Norway, Sweden and Thailand. The subject is very relevant in 2021.
Generally, the paper is well-written but could do with a final spell check, I noticed some minor style errors.
I have some comments which I will outline below, noting the line number in my review document:
37 even though you note 'irrespective of the causes', the majority of incidents reported in the Red Cross report seem to stem from climate change. You could address this, since it's good explanation for the gradual rise in disasters.
Response: Thank you for recognizing this work
We also seized the opportunity to go over the manuscript for brevity, clarity, and consistency, allowing a native English speaker academic perform the necessary English revision.
Point 1 Even though you note 'irrespective of the causes', the majority of incidents reported in the Red Cross report seem to stem from climate change. You could address this, since it's good explanation for the gradual rise in disasters.
Response 1: Thank you for the suggestion. We see the point of the reviewer that climate change is one of the major concerns worldwide. Therefore, we added some information and references to expand the topic of incline numbers of disasters as followed. (Changed in the manuscript line 40-43)
The rate of major incidents and disasters (MID) irrespective of the causes has gradually risen over the past two decades. A major proportion of these incidents are triggered by natural hazards as a result of climate changes and can result in potentially deadly consequences (1).
Point 2 The description of FSC could be structured more, it is not entirely clear now and some sentences seem to be placed here haphazardly, for example ' In previous studies, alternative leadership within the concept of FSC has been discussed (10,13).' does not logically flow in the narrative.
Response 2: Thank you for the suggestion. The definition of surge capacity and its elements are already in lines 49-50. Additional text explaining FSC is added as below, lines 53-64.
In a critical situation, hospitals will try reallocating patients or medical equipment (like ventilators), conducting primary and secondary surge capacity, but the overflow of patients might still outstrip their ability and result in unpredictable consequences, as illustrated in northern Italy during the Covid-19 pandemic in 2020 (9). Therefore, the amplification of MID consequences requires another effort of surge capacity, i.e., “flexible surge capacity” (FSC). FSC aims at scaling up and down all viable resources in the community in terms of all four elements of surge capacity, mentioned above (10–12). As an example, in a previous study, using emergency physicians as alternative leadership in management of MID was discussed within the concept of FSC (10,13).
Point 3 When introducing ACF please describe in more detail what this is. Are they physical sites? They are described like they are but it's not entirely clear to the reader.
Response 3: Thank you for the suggestion. We agree with the reviewer that there were too little details on ACF, referring to an earlier study. Therefore, we modified the contents as followed. (Changed in the manuscript line 65-69)
However, there are limited studies investigating the use of Alternative Care Facilities (ACF), which are places that potentially can be modified into treatment stations for disaster related patients or non-disasters patients, to incorporate the structural needs during an MID as the FSC (15,16).
Point 4 It is not clear to me what you did with the 'Other facilities of interest; veterinary clinics, hotels, and sport facilities'. Did you find these on Google and send them a questionnaire?
Response 4: Thank you for the comment. We shared the concerns of the reviewer about more method explanation, so we addressed the details as followed. (Changed in the manuscript line 93-95)
Other facilities of interest; veterinary clinics and hotels were sampled using an online search through google map. For sport clinics, the information was collected from Bangkok sport arena website online and the consent were made with the administrator in sports division, cultural sports, and tourism department.
Point 5 you mention excluding schools with less than 500 students, did you have a cutoff for these other categories?
Response 5: Thank you for your comment. The cutoff criteria for school had a point to consider because the small schools have limited spaces for casualty collection and treatment. Other categories did not have such limitation. This is clarified in the new version. (Changed in the manuscript line 238-240)
Because of the requirements for FSC, only institutions with significant staff and space larger than 3000 square meters, and with more than 60 employees were included from the entire set of respondents (n=70).
Point 6 to who at the ACF was this questionnaire sent? The answers could be very different based on the function of whoever fills in this questionnaire.
Response 6: Thank you for the comment. We agree with the reviewer that we should address more details on ACFs’ respondents, so we improved as followed. (Changed in the manuscript line 101-103)
All aforementioned facilities received the questionnaire. The ministries, department and responsible persons in each facility were contacted by main author to decide whether the questionnaire should be distributed centrally or by sending to each individual entity.
Point 7 The description of completion is unclear - do you mean that of those that filled in (e.g., 13.8 of hotels) they only filled in half the questionnaires?
Response 7: Thank you for the comment. We understand the concern of the reviewer that there were unclear on the description of forms’ completion. We intended to explain the level of ACFs’ participation; thus, we modified the sentences into more reasonable description as followed. (Changed in the manuscript line 133-136)
From the sample of 967 names and addresses of the facilities, 228 responses (23.6%), were collected through Google forms. Participants who replied to the questionnaire answered all questions. However, the response rate varied in different groups.
Point 8 **Starting at 3.2 ACF specific results: please start the subheading with a short description of how these facilities work in Thailand/BKK, I am a bit confused as to why some of the amounts are so low (please see below).
Response 8: Thank you for the comment. We agree with the reviewer that some facilities in BKK might not have the general performances as others. We added the details as followed. (Changed in the manuscript 171-177)
Line 171-177: the public primary health care centers (PHCC), governed by the Ministry of Health, have doctors, nurses, and pharmacists. The PHCCs provide routine treatments for patients with chronic diseases and minorly acute illnesses. Private clinics, which operate independently yet, require a license from the Ministry of Health have a doctor, and either a nurse or assistant nurse. Most private clinics are specialty clinics providing care for specific niches like pediatrics allergies, or dermatologic issues.
Line 204-205: All dental clinics in Bangkok represented private practices and provided wide ranges of dental procedures.
Point 9 Does this mean that only 12 out of 35 'public primary health care center' felt they could 'stabilize serious injured patients' ? How are these centers organized? Why can only a few 'primary health care center' actually take care of patients? Considering the name I would expect more capabilities. The same goes for private clinics.
Response 9: Thank you for the comment. We understand the reviewer concerns on the capabilities; nonetheless, the information we collected represented the willingness to offer their services, which clearly does not match their real abilities. We included the issue into the discussion, as their limitation were about knowledge and resource insufficiency as in manuscript line 304-308 and 401-405.
Although the number of responses was limited, the study showed that the majority of participants were willing to alleviate the burdens of MIDs. These participants represent the facilities’ leadership. The choice of respondents is based on the assumption of that the leadership has an overview of organizational capacities and ability to act as the voice of the staff(29).
Point 10 (5. Limitations): obviously there is a huge difference between a city the size of my country and 'non-urban settings'. Could you think of more limitations?
Response 10: Thank you for the comment. We agree that the context of the community is the obvious barrier to generalize the outcome. The limitation section is however expanded with other possible limitations (please refer to limitation)
Point 11 Do you have any ideas as to why response rate was low? How could you improve this?
Response 11: Thank you for the comment. We agree with the reviewer and add more details in limitation section as followed. (Changed in the manuscript line 392-400)
One reason for the low participation rate might be the perception of disasters as being rare events, as the awareness of MIDs are low among the Thai population, government officials and business stakeholders. Another reason for low participates might be the fact that disaster management is based on both medical and non-medical measures. There might be a limitation of understanding between organizations involved, i.e., non-medical institutions having limited knowledge about the limitations and possibilities of medical institutions and vice versa (24). In this study, top authorities were contacted to improve the response rate with low success. However, the response rate is concordant with previous non-pandemic studies. In future study, electronic repeated reminders might improve response rates (40–42).

Reviewer 5 Report
Thank you for the opportunity to review your manuscript. The study addresses a highly topical subject of increasing importance. However, I found the manuscript quite descriptive and I have a number of concerns, particularly around the methods, results and discussion sections, and overall conclusions drawn. Overall, I felt the article has merit but requires a number of significant changes to improve its readability, rigor and interest to the journal's readership.
- All sections of the manuscript, including the title and abstract, require a substantial amount of editing and proofing. There are a large number of grammatical and syntax errors. A number of sections are consequently quite difficult to understand. It was only upon reading the article by Glantz, et al. (reference 11), on which this study was based, that the background and purpose of the manuscript became clear.
- In the methods section, it would be helpful to describe the study design. This would strengthen the justification for the chosen approach and clarify how samples were selected, and data was collected, analysed and interpreted.
- The data processing section requires more detail. It was unclear how quantitative data from the surveys were analysed. It was also unclear how qualitative survey data and interview data were analysed - were they analysed separately or together using an inductive or deductive approach? How were codes decided upon, assigned and themes identified?
- In the results section, it was very unclear whether the data/findings reported related to survey or interview data. This was particularly confusing as the authors referred to "all/few/some facilities" but didn't specify whether this was all/few/some facilities that responded to the survey or participated in the interview.
- The results section was quite descriptive. The quantitative data described in sections 3.2.2 to 3.2.6 may be more concisely presented as a table.
- I have some concerns about over-interpretation of the data in the discussion section. For example, the authors state (lines 250-251) that the study positively affirms that facilities are able to... partake in response to an MID but the data presented only suggests that facilities may be willing to partake in a response. Similarly, how do the study authors know that a significant number of staff in the investigated facilities of interest would be willing to partake in the response system (lines 262-263)? Were all staff from each facility surveyed? How was capacity and capability (line 281) determined? The data presented was largely self-reported perceptual data derived from a single respondent per facility in response to a hypothetical situation. Ability, capacity and capability to partake in an FSC response were not directly measured nor tested. This study limitation should be discussed in section 5.
- The first sentence of the conclusion is a little awkward perhaps should be softened. I'm not sure what the authors mean by applicable - do they mean potentially feasible given that facilities appear willing to participate in an FSC?
Author Response
Reviewer: Thank you for the opportunity to review your manuscript. The study addresses a highly topical subject of increasing importance. However, I found the manuscript quite descriptive, and I have a number of concerns, particularly around the methods, results and discussion sections, and overall conclusions drawn. Overall, I felt the article has merit but requires a number of significant changes to improve its readability, rigor and interest to the journal's readership.
We seized the opportunity to go over the manuscript for brevity, clarity, and consistency, allowing a native English speaker academic perform the necessary English revision.
Point 1: All sections of the manuscript, including the title and abstract, require a substantial amount of editing and proofing. There are a large number of grammatical and syntax errors. A number of sections are consequently quite difficult to understand. It was only upon reading the article by Glantz, et al. (reference 11), on which this study was based, that the background and purpose of the manuscript became clear.
Response 1: Thank you for the comment. We shared the reviewer concerned and addressed the extensive English revision by letting an academic native English speaker review and edit the paper before re-submission.
Point 2: In the methods section, it would be helpful to describe the study design. This would strengthen the justification for the chosen approach and clarify how samples were selected, and data was collected, analysed and interpreted.
Response 2: Thank you for the comment. We understand your concerns and expand the methodology section.
ACFs facilities: The names and location of public primary healthcare centers, private clinics, and dental clinics were obtained from the Ministry of Health in Bangkok. The names of schools were obtained from the department of education. Veterinary clinics, hotel and sport arenas were searced for through online and available websites.
Sampling: All aforementioned facilities received the questionnaire. The ministries, department and responsible persons in each facility were contacted by main author to decide whether the questionnaire should be distributed centrally or by sending to each individual entity.
Sampling. The sampling for government-related institutions was done by the Ministry of Health and Department of Education officials and the authors had no influence on the selection. For other facilities, all available facilities were contacted and received the questionnaire. However, the authors had no influence on the response from these facilities, and despite multiple contacts, only those with complete responses were included.
For data collection and analysis, please see next question.
Point 3: The data processing section requires more detail. It was unclear how quantitative data from the surveys were analysed. It was also unclear how qualitative survey data and interview data were analysed - were they analysed separately or together using an inductive or deductive approach? How were codes decided upon, assigned and themes identified?
Response 3: Thank you for the comment. We agree with the reviewer that the quantitative and qualitative data analyses could be explained in more details, and we addressed those details as followed. (Changed in the manuscript line 104-131)
The Questionnaire (Appendix A): An already published and validated questionnaire (Cronbach’s alpha with an internal consistency of 0.739(18)) was utilized(11). It was translated into Thai and then back into English to assure the accuracy of the questions. The face validation of the original questionnaire was based on logic, relevance, comprehension, legibility, clarity, usability, and consensus. The questionnaires referred to a situation that the facilities of interest faced, a fictitious scenario of a mass casualty incident. ACFs were asked about the care they could provide to the healthcare actors in the area.
There were both open-ended and close-ended questions to capture and generate relevant data, and the answers were quantitatively collected (19,20). The quantitative data dealt with the number of participants, who received the questionnaire and the number of those responding and thus, the rate of the participation. No other quantitative data was presented.
The qualitative data was collected by semi-constructive interviews, which in contrast to a structured interview technique, allows the participant to divert and suggest new ideas during the interview based on responses. The interviewer in a semi-structured interview generally has a framework of themes to be explored (21). All data was recorded and analyzed by the first investigator.
The theme applied to the interview was the one used in Glantz’s study (2020), who used the same questionnaire and interview questions. For analyzing, we deductively used the distribution of the concepts of surge capacity, which allows for examining communication by using text directly. This method allows for both qualitative and quantitative analysis, if needed. It is also considered a relatively exact research method if it is done correctly (limitation). However, it is an inexpensive research method and considered a more powerful tool when combined with other research methods such as interviews. However, it has also its limitations (see limitation). Thematic content coding was performed to identify competencies, challenges, and interest to take part in the FSC-response system. No more interviews were held after reaching the point of data saturation (20).
Limitation: It is subject to increased error, particularly when relational analysis is used to attain a higher level of interpretation. However, the main investigator was part of a similar study published 2020 and had experience of using it.
Point 4: In the results section, it was very unclear whether the data/findings reported related to survey or interview data. This was particularly confusing as the authors referred to "all/few/some facilities" but didn't specify whether this was all/few/some facilities that responded to the survey or participated in the interview.
Response 4: Thank you for the comment. We see the point of the reviewer and we stressed the interview results as followed. (Changed in the manuscript line 191-201)
In addition, more than half of the respondents (21 public centers and 10 private clinics) commented that they lacked essential medical equipment including advanced defibrillators, intubation equipment and resuscitation medications. Moreover, increased space, extra ambulance vehicles and more emergency kits would ease and encourage their contribution. The in-depth interviews highlighted this insufficiency in both public and private healthcare facilities. Although all public facilities had doctors and nurses, and some centers had pharmacists to include in a potential FSC, the qualitative data collection revealed barriers in private clinics’ willingness to contribute doctors or nursing assistants. Furthermore, many respondents requested training in advanced life support, emergency and trauma case management, disaster management and prehospital transportation in their interviews. Though most respondents offered constructive criticism, a limited number of answers (3 public clinics and 5 private clinics) responded negatively toward any kind of involvement.
Point 5: The results section was quite descriptive. The quantitative data described in sections 3.2.2 to 3.2.6 may be more concisely presented as a table.
Response 5: Thank you for the comment. We agree with the reviewer and add the quantitative data as in table 2 and 3
Point 6: I have some concerns about over-interpretation of the data in the discussion section. For example, the authors state (lines 250-251) that the study positively affirms that facilities are able to... partake in response to an MID but the data presented only suggests that facilities may be willing to partake in a response. Similarly, how do the study authors know that a significant number of staff in the investigated facilities of interest would be willing to partake in the response system (lines 262-263)? Were all staff from each facility surveyed? How was capacity and capability (line 281) determined? The data presented was largely self-reported perceptual data derived from a single respondent per facility in response to a hypothetical situation. Ability, capacity and capability to partake in an FSC response were not directly measured nor tested. This study limitation should be discussed in section 5.
Response 6: Thank you for the comment. We agree with the over-state of the wording, and we changed them as followed: Line 307-309: The choice of respondents is based on the assumption of that the leadership has an overview of organizational capacities and ability to act as the voice of the staff(29). The four components of surge capacity will be discussed.
Point 7: The first sentence of the conclusion is a little awkward perhaps should be softened. I'm not sure what the authors mean by applicable - do they mean potentially feasible given that facilities appear willing to participate in an FSC?
Response 7: Thank you for the comment. We agree with the reviewer and modify the conclusion as followed. (Changed in the manuscript line 407-415)
The concept of an FSC response system is applicable to the metropolitan area of this study where several clinics, schools and hotels exist. New studies will reveal if the apprehensive points could be generalized to other countries with the same context. However, the development and implementation of educational initiatives including, exercises, drills, and maintenance courses and financial contingency were recognized to be barrier necessary to overcome in implementation of the FSC concept. The success of overcoming these barriers would enable sustainable FSC systems to be developed within the MID response system.

Round 2
Author Response
Thank you for your evaluation of the revised manuscript.
Reviewer 5 Report
Thank you for the opportunity to review a much-improved revised manuscript. I found the description of the study purpose, design clearer and easier to understand. The presentation and interpretation of study findings was also improved. I have a few minor suggested corrections:
line 65 - "covid" should be "COVID"
line 86 - "governmentsal" should be "governmental" or "government"
lines 170-171 - delete "Their comments were extensive and critical".
line 188 - "cares" should be "care"
line 206 - "minorly" should perhaps be "minor", which is more commonly used
Table 2 - suggest identifying this as survey/questionnaire data in the table title. Also suggest that you include the sample size for each respondent group in the top line as per table 3.
Table 3 - as per above, suggest identifying this as survey data in table title.
line 479 - delete "assigned to them"
line 524 - delete "troubled"
line 535 - unclear how the authors determined that private clinics had an unclear line of command. Did they meant that it was difficult to identify the senior managers in these organizations from publicly available information?
lines 550-552 - suggest rewording to "Therefore, exercises, training, and practice guidelines may empower both public and private organizations to learn how to work together in emergencies like hospital evacuation."
line 594 - suggest replacing "same" with "similar".
Author Response
Response to Reviewer 5 comments
Thank you for your evaluation of the revised manuscript.
Point 1: line 65 – “covid” should be “COVID”
Response1: Thank you for your comment. Done
Point 2: line 86 – “governmentsal” should be “governmental” or “government”
Response 2: Thank you for your comment. Done.
Point 3: lines 170-171 - delete "Their comments were extensive and critical".
Response 3: Thank you for your comment. Done.
Point 4: line 188 - "cares" should be "care"
Response 4: Thank you for your comment. Done.
Point 5: line 206 - "minorly" should perhaps be "minor", which is more commonly used
Response 5: Thank you for your comment. Done.
Point 6: Table 2 - suggest identifying this as survey/questionnaire data in the table title. Also suggest that you include the sample size for each respondent group in the top line as per table 3.
Response 6: Thank you for your comment. We agree with your suggestion and change it in the manuscript.
Point 7: Table 3 - as per above, suggest identifying this as survey data in table title.
Response 7: Thank you for your comment. We agree with your suggestion and change it in the manuscript.
Point 8: line 479 - delete "assigned to them"
Response 8: Thank you for your comment. Done
Point 9: line 524 - delete "troubled"
Response 9: Thank you for your comment. Done
Point 10: line 535 - unclear how the authors determined that private clinics had an unclear line of command. Did they meant that it was difficult to identify the senior managers in these organizations from publicly available information?
Response 10: Thank you for your comments. We understand the reviewer point. The private clinics are independently operated which explained in the result part so it can be interpreted that they have no line of command. However, we clarify the sentences as below.
and they have an unclear line of command because they are independently governed and the Ministry of Health has no power over them.
Point 11: lines 550-552 - suggest rewording to "Therefore, exercises, training, and practice guidelines may empower both public and private organizations to learn how to work together in emergencies like hospital evacuation."
Response 11: Thank you for your comments. We see the reviewer point and changed in the manuscript.
Point 12: line 594 - suggest replacing "same" with "similar".
Response 12: Thank you for your comment. Done.
